# Early life factors associated with childhood trajectories of violence among the Birth to Twenty-Plus Cohort in Soweto, South Africa

Lilian Muchai[1]*, Sara Naicker[2,3], Juliana Kagura[1]

**1** School of Public Health, Faculty of Health Sciences, University of the Witwatersrand, Johannesburg, Gauteng, South Africa, **2** Research, Development, Science and Innovation Unit, Human Sciences Research Council, Pretoria, South Africa, **3** DSI-NRF Centre of Excellence in Human Development, University of the Witwatersrand, Johannesburg, Gauteng, South Africa

These authors contributted equally to this work.
* muchailily@gmail.com

## Abstract

Violence against children (VAC) has devastating and long-term negative consequences on health, social and economic well-being at both the individual and societal levels. There is limited research on the life course experience of VAC, especially in Africa. This study aimed to identify trajectories of physical and sexual violence victimization in childhood and evaluate early life factors predicting these violence trajectories. This study used data from birth to 18 years from the ongoing prospective Birth to Twenty Plus cohort (Bt20+). Analyses included children who reported experiences (yes/no) of physical and sexual violence at a minimum of two time points between five and 18 years. Group-based trajectory modelling was employed to identify groups of children with similar patterns of violence over time, while multivariable logistic regression was used to establish early life factors associated with violence trajectory group membership. Separately, two trajectory groups of physical violence (*adolescent limited* (65.1%) and *chronic increasing* (34.9%)) and sexual violence (*adolescent limited* (74.1%) and *late increasing* (25.9%)) victimization were identified. Early life factors associated with a higher risk of *chronic increasing* trajectory group membership, after adjusting for covariates, were being male (adjusted odds ratio [aOR] 1.67, 95% CI 1.31; 2.10) and having a mother with at least secondary education compared to higher education (aOR 1.73, 95% CI 1.08; 2.76). In addition, residing in middle, compared to low, socioeconomic households (aOR 0.68, 95% CI 0.50; 0.92) was protective against membership in this group. Residing in high compared to low socioeconomic households, was the only early life factor whose association approached significance with membership in the *late-increasing* sexual violence victimization trajectory group (aOR 0.63, 95% CI 0.42; 0.95). In conclusion, children follow different violence victimization trajectories across their childhoods. Identifying early life factors predicting violence trajectories provides targeted prevention intervention areas that can mitigate children's violence experience.

**Data availability statement:** The Birth to Twenty- Plus Cohort data can be requested from the Birth to Thirty Executive Committee: Prof. Linda Ritcher, the executive committee chair, linda.richter@wits.ac.za. The first five years of cohort data are available online at bt30.org (https://bt30.org/data/ ).

**Funding:** The Bt20+ cohort study has been supported by the Wellcome Trust (UK), the South African Medical Research Council, the Bill and Melinda Gates Foundation, the DSI-NRF Centre of Excellence in Human Development at the University of the Witwatersrand, and the Developmental Pathways for Health Research Unit. This work was funded by the Sexual Violence Research Initiative. The funders had no role in the study design, analysis, decision to publish or preparation of the manuscript.

**Competing interests:** The authors have declared that no competing interest exist.

## Introduction

Globally, more than half of children between the ages of 2 and 17 years reported having experienced some type of violence each year, according to a systematic review of global violence prevalence [1]. Africa bears the biggest brunt of violence against children (VAC), with the prevalence of physical violence at 60% and 51% for boys and girls, respectively [2]. Violence is endemic to South Africa, particularly associated with structural and systemic violence originating from the Apartheid period. A nationally representative survey among children 15–17 years of age reported that almost 20% of children had experienced some form of sexual abuse, 21.3% experienced neglect, and over 30% and 16% experienced physical and emotional abuse, respectively [3]. Children are likely to experience different forms of violence at different stages of their lives, as their environments and interactions change [4]. This ranges from violent discipline during early childhood by caregivers, bullying by peers at school, sexual violence victimization and intimate partner violence during the adolescent period – although the boundaries of these timeframes are porous [5,6]. The prospective Birth to Twenty Plus study in Soweto found that almost 40% of children were exposed to up to six types of violence throughout their lives, including exposure to violence in the community, home and school; exposure to peer violence; being a victim of physical violence, and sexual violence [7].

Despite mounting evidence of the determinants and the burden of VAC, few studies have demonstrated the effects of life course exposure to multiple forms of violence, and even fewer studies have identified the different clusters of violence patterns (trajectories) over time. Previous studies, conducted outside of Africa, have demonstrated heterogeneity in the number and shapes of trajectories of physical and/ or sexual violence victimization, with two [8,9] three [8,10], four [11] and five [12] trajectory groups identified. Most of these studies assess trajectories of violence victimization as predictors of subsequent physical and mental health outcomes in adulthood, rather than the identification of risk factors for trajectory group membership. And where they do, most studies assessing risk factors for violence trajectory group membership in childhood tend to focus on violent aggression or perpetration. However, a significant overlap between victimization and perpetration experiences is documented in research [13–16], showing that victims and perpetrators share similar contextual features that place them at an increased risk for violence victimization and perpetration.

The developmental origins of health and disease (DOHaD) perspective stipulates that experiences or exposures during sensitive developmental period – specifically during utero and early childhood – can have negative long-term consequences in the risk for poor health, disease and behavioural problems throughout the life-course [17,18]. Several demographic and early life factors have been associated with violence trajectories. Being male, residing in households with low socioeconomic status, early motherhood, having more than one child in the household, single parenthood and low parental education were some of the risk factors identified for membership in persistent physical violence victimization or aggression trajectory groups [11,19–21]. From cross-sectional analyses conducted in different countries, early life factors

predicting violence victimization include household crowding [22], paternal absence [23] and mother's prior history of violence [24,25]. Factors along the pathway of violence experiences have also been identified. Both low [26] and high birth weight [27,28], and catch-up growth (weight and length) in infancy are documented to predict future overweight and obesity [29,30], delayed cognitive development [31] and poor academic performance [29], which are risk factors for violence victimization and perpetration [32–34].

There is a dearth of studies conducted in Africa assessing factors associated with childhood trajectories of violence due to a lack of longitudinal data on violence. This limits the ability to assess the developmental course of violence and the heterogeneity of violence trajectories in populations outside of the Global North. This is paramount towards the identification, development and implementation of context-specific prevention strategies prior to the child's exposure or experience of violence. The current study aims to generate physical and sexual violence victimization trajectories across childhood and assess which factors in early life predict membership into specific violence trajectories.

## Materials and methods

### Study setting and site

The primary study (Birth to Twenty plus study, Bt20+) is a prospective birth cohort study of singleton children born between April 23 and June 8, 1990 in public health facilities in Soweto, South Africa. Soweto is a densely populated suburb located within the Johannesburg metropolitan municipality in the Gauteng province in South Africa. The primary study included pregnant women attending public antenatal clinics with expected delivery dates within the above-mentioned dates, and later continued residence of the mother and the baby within the study area during the child's first six months of life. Data collected in the Bt20 + study are multidisciplinary, tracking infant, child, adolescent and now adult physical, social and psychosocial domains [35]. Of the 3,273 children enrolled at study inception, follow-up has been conducted annually or bi-annually through home visits, at the study sites or through school surveys. Full descriptions of the Bt20 + cohort, its attrition and methods have been published elsewhere [35,36].

### Study design

This study involved secondary data analysis of the prospective Bt20 + cohort study. Data from birth until 18 years (1990–2018) from the primary study were included in this secondary study. The study design is illustrated in Fig 1 below.

### Study population and sample

The study population included children from birth to 18 years from the primary prospective Bt20 + cohort study. The cohort is semi-representative of the South African population, with a higher percentage of Black Africans and a slight under-representation of White families [35].

From the 19 waves of data collection between birth to 18 years, four time points (years five, 11, 15 and 18) had sufficient data on physical and sexual violence victimization, with each violence type measured separately. Children in the study population who reported any or no experiences of physical and sexual violence in at least two of the four time points were included in this study.

### Study variables

**Outcome variables.** Two outcome variables – physical and sexual violence victimization – were considered for this study. In early childhood, from age five to 10 years, the primary caregiver reported on the experiences of the child, and from age 11 onwards, experiences were self-reported.

Physical violence victimization was coded as a binary variable (yes/no) generated from questions on any experience of physical violence in the previous 12 months based on the following four categories: at home, at school, in the

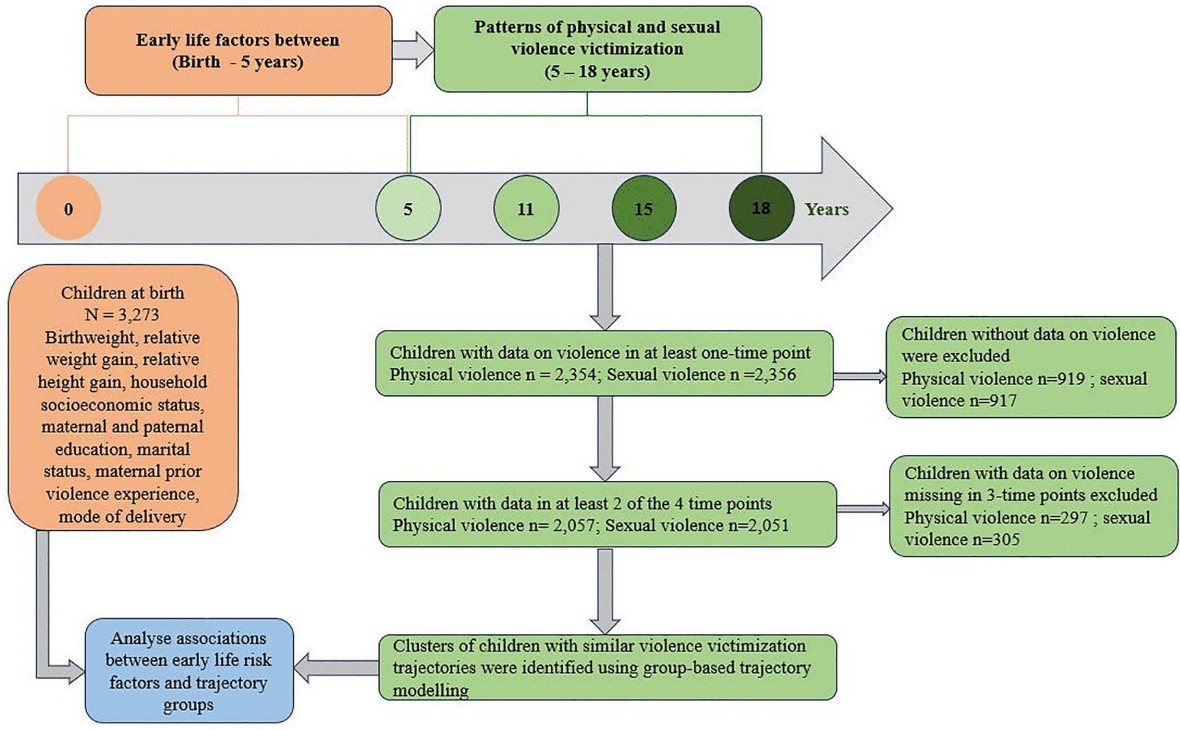

**Fig 1. Study design.**

neighbourhood, and at work (only year 18). The questions included: have you been (a) physically hurt; (b) hit;(c) kicked; (d) badly beaten up; (e) attacked with a knife or sharp object. Responses to each of these questions were never, once or twice, few times, and many times. The responses were recoded to a dichotomous variable with responses no, coded '0' (never) and yes, coded '1' (once or twice, few times and many times combined). A positive response to any of the five questions for each of the four categories mentioned above was coded as having been a victim of physical violence (yes) for that specific time point of data collection.

Similarly, sexual violence victimization was coded as a binary outcome of sexual violence experience in the preceding 12 months based at home, at school, in the neighbourhood, and at work. Sexual violence in the primary study was defined as unwanted sexual experiences. Questions included: have you been (a) sexually assaulted/attacked (b) sexually harassed (c) wanted to engage in oral sex (d) wanted to have oral sex last month (e) wanted to have sex (f) wanted to have sex last month (g) wanted to engage in heavy petting or foreplay. Responses to questions a and b were never, once or twice, a few times or many times. These were recoded to no, coded '0' (never) and yes, coded '1' (once or twice, a few times and many times). Yes or no responses were recorded for questions c, d, e, f and g. Questions c to g were limited to children who had engaged in sexual activities. These questions were reverse coded, where a no response, for example, that the individual did not want to engage in sex was recoded yes, coded '1' to having experienced sexual violence. A positive response to any of the questions above was coded as having been a victim of sexual violence (yes).

For details on the proportions of physical and sexual violence across the given time points refer to Table in S1 Table.

**Risk factors.** Early life factors measured between birth and five years associated with violence experience were selected based on their significance in literature and data availability. These selected covariates for physical and sexual violence victimization trajectories were grouped into individual and family level factors.

*Individual/Child-level factors*: Birth weight was obtained from birth notifications and categorized as low birth weight (less than 2,500 grams) or normal birth weight (2,500 grams or higher). Relative weight gain and relative height gain were derived as standardized residuals computed from sex-specific linear regression of these growth changes measured between infancy (0–2) and early childhood (2–5 years).

*Family level factors*: Household socioeconomic status was based on total household asset scores, grouped into tertiles of low (coded '1'), middle (coded '2') and high (coded '3') socioeconomic status. Examples of some of the household assets included possession of a television, refrigerator, washing machine and a car. Household crowding was defined as households whose ratio of persons per sleeping room was equal to or above the total sample household crowding ratio mean, and coded '1' for crowding and '0' for no crowding. Maternal age was recorded at birth and grouped for analysis – 24 years and below (coded '0'), between 25 and 34 years (coded '1') and above 35 years (coded '2') at the time of birth. Parity was coded as only child from mother (coded '0') or mother had more than one child (coded '1'). Maternal and paternal education were categorized as primary school and below (coded '0'), secondary school (coded '1'), and post-school training (diploma, bachelors, masters or doctoral degrees, coded '2'). Marital status was recorded as married, living together, widowed/ divorced or single. This was recoded to married (married or living together, coded '1') and single (widowed/divorced or single, coded '0'). Father's presence in the household was coded as yes ('1') or no ('0'). Maternal prior violence experience was derived using data from two variables. Any positive (yes) response on either mother's prior violence in childhood or intimate partner violence during pregnancy was coded as mother having experienced violence (yes, coded '1'). Finally, the mode of delivery obtained from birth notifications was recoded to normal vaginal delivery ('0') or assisted (caesarean section/ use of forceps or vacuum) delivery ('1').

## Data management and analysis

De-identified electronic data from the Birth to Twenty Plus study database was accessed on the 20th of April 2023 following the ethical approval of this research. These data were imported into Stata 17 for cleaning and analysis. Only data on early life factors between the antenatal period to five years of age and data on physical and sexual violence victimization between ages five to 18 years were extracted from the primary Bt20＋database. The data was checked for missing and duplicate values, followed by recoding and generation of new variables. No duplicates were identified and the final study dataset was stored in a password-protected file for analyses.

**Identification of trajectory groups.** Group based trajectory modelling (GBTM) was used to identify clusters of children with similar violence victimization trajectories from age five to 18 years. Group based trajectory modelling is a semiparametric group strategy that applies finite mixture models to identify clusters of persons (trajectory groups) with similar developmental course of an outcome over time [37,38]. This technique assumes that the population is composed of distinct groups defined by their outcome progression over time and makes no assumptions about the population distribution of the trajectories [37]. Determination of the number of trajectory groups that best fit the data was based on the most parsimonious model determined by lowest Bayesian Information Criterion (BIC) and high entropy after comparison of two and three-group models. Linear, quadratic and cubic polynomial orders were tested to identify the polynomial order that best characterized the evolution of violence victimization over time. The logit function reflecting the Bernoulli distribution of the outcome was specified in all the models tested. The final model was selected based on theoretical plausibility, average posterior probabilities of trajectory group membership greater than the set threshold of 0.70, highest entropy and group membership greater than 10% [37]. All trajectory analyses were conducted using the Stata plug-in Traj [39].

**Early life factor covariates for trajectory membership.** Descriptive analyses were conducted for the individual and family level early life factors disaggregated by sex. Participants were assigned to the trajectory group with the maximum probability of group membership. A two-step process was used to identify early life factors associated with childhood trajectories of physical and sexual violence victimization. First, the independent association of each covariate with

trajectory group membership was assessed using univariable logistic regression analysis. Second, covariates for inclusion in the multivariable model were selected based on documented associations with violence victimization and the availability of sufficient data; all covariates meeting these criteria were retained in the final adjusted model. Odds ratios, 95% confidence intervals and p-values were reported. P-values <0.05 were considered statistically significant in the univariable and multivariable models. Post regression diagnostics were conducted to ensure the validity and robustness of the final model. The test for multicollinearity was undertaken to evaluate interdependence between explanatory variables, and the final model adequacy was assessed using the Hosmer -Lemeshow goodness of fit test, which evaluates the agreement between the observed outcome and the model estimated probabilities.

### Ethical considerations

Ethical approval for the primary Bt20+study was obtained from the Human Ethics and Research Committee (HREC-Medical) of the University of Witwatersrand prior to the start of the study (certificate number M111182). Written informed consent was initially obtained from the primary caregivers of the participants. At the appropriate age, both children and the caregivers then provided written informed consent. For this secondary data analyses, permission to use the data was granted by Development Pathways for Health Research Unit (DPHRU) responsible for the primary study and a memorandum of agreement was signed. Ethical clearance for the current study was granted by HREC-Medical at the University of Witwatersrand (certificate number M230218). Confidentiality of data was maintained throughout the research period by use of anonymized data which was password protected and backed up for safety.

## Results

### Study characteristics

A total of 3,273 children were enrolled at birth. Of these, 2,057 (62.9%) reported experiences of physical violence victimization and 2,051 (62.7%) reported sexual violence victimization in at least two of the four time points of data collection, and were included in the study. Experience of physical violence ranged from 14.0% at age five, with a peak of 71.8% at age 15, which declined to 38.6% at 18 years of age. Sexual violence experiences ranged from 0.8% at five years to a peak of 28.9% at the age of 18 years (Table in S1 Table). A total of 1,212 (37.0%) and 1,218 (37.2%) participants did not meet the inclusion criteria and were excluded from the analyses for physical and sexual violence victimization trajectories, respectively. For both analyses, there were no significant differences between included and excluded samples with reference to sex, maternal age at birth, father's presence and birth weight. Compared to children excluded in this study, a greater proportion of children included in the study resided in poorer households, which were crowded and headed by a single parent. In addition, the included sample had a lower proportion of parents with post school training compared to children in the excluded sample (Tables in S2 and S3 Table). Covariates for physical and sexual violence victimization trajectories that were significantly different between included and excluded samples, were adjusted for in the multivariable analyses.

**Study characteristics by sex.** Among the 2,057 children included in the physical violence and 2,051 children included in the sexual violence victimization trajectory analyses, there was a higher proportion of girls (52.5% physical violence; 52.4% sexual violence) than boys (Table 1). More girls (12.7%) than boys (9.7%) were born with low birth weight. Overall, a higher proportion of children were born in households with low socioeconomic status (65.0% physical violence; 64.8% sexual violence), had one or more siblings (61.9% physical violence; 61.8% sexual violence) and resided in a single parent household at birth (62.0%). A higher proportion of fathers (18.8% physical violence; 18.7% sexual violence) compared to mothers (8.6%) had attained higher schooling after completion of secondary school in both analyses.

### Trajectories of physical and sexual violence victimization

Following the examination of different trajectory groups and polynomial functions, a two-group model was selected as the most parsimonious model that best described patterns for both physical and sexual violence victimization between

**Table 1. Description of study sample characteristics stratified by sex.**

| Variables | Physical violence victimization | | | Sexual violence victimization | | |
|---|---|---|---|---|---|---|
| | Total N (%) | Female n (%) | Male n (%) | Total N (%) | Female n (%) | Male n (%) |
| Total sample | 2057 | 1080 (52.5) | 977 (47.5) | 2051 | 1074 (52.4) | 977 (47.6) |
| **Outcome variables** | | | | | | |
| **Violence trajectories** | | | | | | |
| Adolescent limited | 1340 (65.1) | 770 (71.3) | 570 (58.3) | 1519 (74.1) | 801 (74.6) | 718 (73.5) |
| Increasing (chronic/late) | 717 (34.9) | 310 (28.7) | 407 (41.7) | 532 (25.9) | 273 (25.4) | 259 (26.5) |
| **Exposure variables** | | | | | | |
| **Individual level factors** | | | | | | |
| **Birth weight** | | | | | | |
| Low birth weight (<2500 grams) | 231 (11.3) | 137 (12.7) | 94 (9.7) | 230 (11.2) | 136 (12.7) | 94 (9.7) |
| Normal birth weight (≥ 2500 grams) | 1822 (88.8) | 942 (87.3) | 880 (90.4) | 1817 (88.8) | 937 (87.3) | 880 (90.4) |
| **Infant and child growth** | | | | | | |
| Relative weight gain 0–2 years[a] | 1401 | −0.01 (1.00) | −0.05 (1.00) | 1398 | −0.01 (1.00) | −0.05 (0.99) |
| Relative weight gain 2–5 years[a] | 1292 | −0.03 (1.03) | 0.02 (1.03) | 1291 | −0.03 (1.03) | 0.02 (1.03) |
| Relative height gain 0–2 years[a] | 1402 | −0.03 (0.99) | −0.06 (0.98) | 1399 | −0.03 (0.99) | −0.06 (0.98) |
| Relative height gain 2–5 years[a] | 1292 | 0.00 (0.97) | 0.00 (1.03) | 1291 | 0.00 (0.97) | 0.00 (1.03) |
| **Family level factors** | | | | | | |
| **Household socioeconomic status** | | | | | | |
| Low | 1217 (65.0) | 630 (64.3) | 587 (65.8) | 1212 (64.8) | 626 (64.1) | 586 (65.6) |
| Middle | 392 (20.9) | 200 (20.4) | 192 (21.5) | 392 (21.0) | 199 (20.4) | 193 (21.6) |
| High | 263 (14.1) | 150 (15.3) | 113 (12.7) | 266 (14.2) | 152 (15.6) | 114 (12.8) |
| **Household crowding** | | | | | | |
| Yes | 744 (41.8) | 378 (40.3) | 366 (43.5) | 745 (42.0) | 376 (40.3) | 369 (43.8) |
| No | 1034 (58.2) | 559 (59.7) | 475 (56.5) | 1030 (58.0) | 556 (59.7) | 474 (56.2) |
| **Maternal age at the time of birth** | | | | | | |
| ≤ 24 years | 962 (46.8) | 504 (46.8) | 458 (46.9) | 961 (46.9) | 502 (46.8) | 459 (47.0) |
| 25–34 years | 879 (42.8) | 466 (43.2) | 413 (42.3) | 875 (42.7) | 463 (43.2) | 412 (42.2) |
| ≥ 35 years | 214 (10.4) | 108 (10.0) | 106 (10.9) | 213 (10.4) | 107 (10.0) | 106 (10.9) |
| **Maternal parity** | | | | | | |
| 1 child | 783 (38.1) | 423 (39.2) | 360 (36.9) | 784 (38.2) | 423 (39.4) | 361 (37.0) |
| >1 child | 1274 (61.9) | 657 (60.8) | 617 (63.2) | 1267 (61.8) | 651 (60.6) | 616 (63.1) |
| **Marital status** | | | | | | |
| Married | 773 (37.8) | 394 (36.7) | 379 (39.1) | 773 (37.9) | 392 (36.7) | 381 (39.2) |
| Single | 1270 (62.2) | 679 (63.3) | 591 (60.9) | 1266 (62.1) | 676 (63.3) | 590 (60.8) |
| **Maternal education status** | | | | | | |
| Primary & below | 232 (12.3) | 124 (12.5) | 108 (12.1) | 233 (12.4) | 125 (12.6) | 108 (12.1) |
| Secondary | 1491 (79.1) | 786 (79.3) | 705 (78.9) | 1490 (79.1) | 785 (79.4) | 705 (78.8) |
| Post school training | 162 (8.6) | 81 (8.2) | 81 (9.1) | 161 (8.6) | 79 (8.0) | 82 (9.2) |
| **Paternal education status** | | | | | | |
| Primary & below | 121 (8.3) | 67 (8.8) | 54 (7.8) | 120 (8.3) | 66 (8.7) | 54 (7.8) |
| Secondary | 1061 (72.9) | 543 (71.4) | 518 (74.6) | 1060 (73.0) | 542 (71.6) | 518 (74.5) |
| Post-school training | 273 (18.8) | 151 (19.8) | 122 (17.6) | 272 (18.7) | 149 (19.7) | 123 (17.7) |
| **Father present** | | | | | | |
| Yes | 1568 (84.9) | 825 (85.0) | 743 (84.7) | 1564 (84.8) | 822 (84.9) | 742 (84.7) |
| No | 280 (15.2) | 146 (15.0) | 134 (15.3) | 280 (15.2) | 146 (15.1) | 134 (15.3) |

*(Continued)*

**Table 1.** (Continued)

| Variables | Physical violence victimization | | | Sexual violence victimization | | |
|---|---|---|---|---|---|---|
| | Total N (%) | Female n (%) | Male n (%) | Total N (%) | Female n (%) | Male n (%) |
| **Maternal prior violence experience** | | | | | | |
| Yes | 184 (18.4) | 91 (17.3) | 93 (19.5) | 185 (18.4) | 92 (17.5) | 93 (19.5) |
| No | 818 (81.6) | 435 (82.7) | 383 (80.5) | 819 (81.6) | 434 (82.5) | 385 (80.5) |
| **Mode of delivery** | | | | | | |
| Vaginal delivery | 1012 (88.2) | 525 (88.8) | 487 (87.6) | 1006 (88.2) | 521 (88.8) | 485 (87.6) |
| Assisted delivery | 135 (11.8) | 66 (11.2) | 69 (12.4) | 135 (11.8) | 66 (11.2) | 69 (12.5) |

[a]Values are means (standard deviation).

five and 18 years of age. However, the shape or patterns of violence victimization differed between physical and sexual violence victimization trajectories.

**Physical violence victimization trajectories.** Both group trajectories of physical violence victimization had an early onset of violence victimization. These groups were (a) *adolescent limited* (65.1%) characterized by a gradual increase in violence victimization from age five to 15 years, after which there was a decrease in violence victimization, and (b) the *chronic increasing* group (34.9%) characterized by a persistently increasing pattern of violence victimization throughout adolescence (Fig 2). The two-group model selected had the lowest BIC value, with average posterior probabilities for group membership greater than 0.70 for the *adolescent limited* group (0.93) and the *chronic increasing* group (0.73). This model had medium entropy of 0.56 reported (Table in S4 Table).

**Sexual violence victimization trajectories.** Fig 3 below shows the two-group trajectories of sexual violence victimization that were characterized by a late onset of sexual violence victimization. These groups were the (a) *adolescent limited* (74.1%) and (b) *late increasing* (25.9%) trajectory groups of sexual violence victimization. The

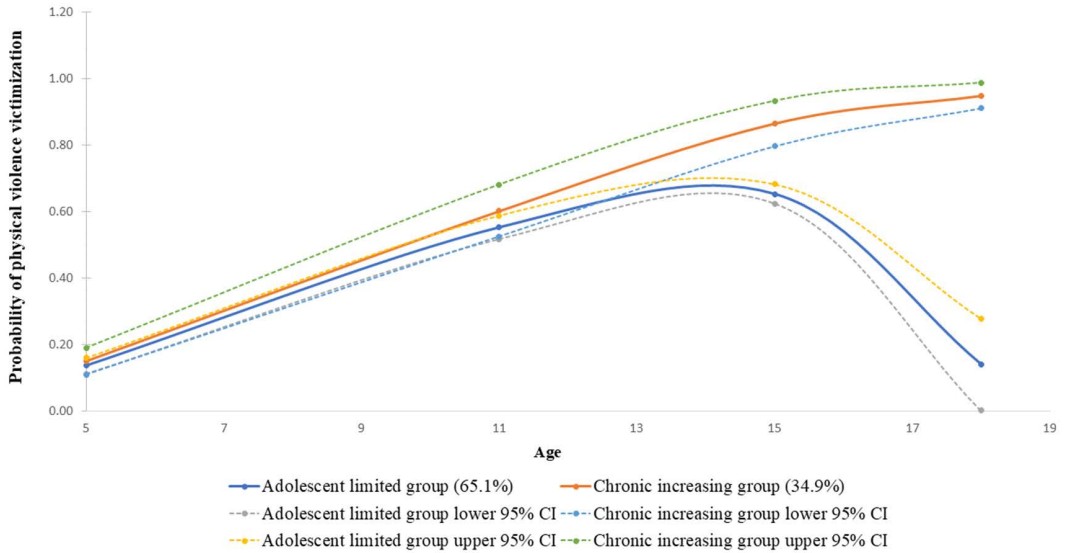

**Fig 2. Physical violence victimization trajectories among children between 5 and 18 years of age.** Note; lines show estimated trajectories, points at each age represent the observed proportion of children between the age of 5 to 18 years reporting physical violence victimization according to the assigned physical violence trajectory group.

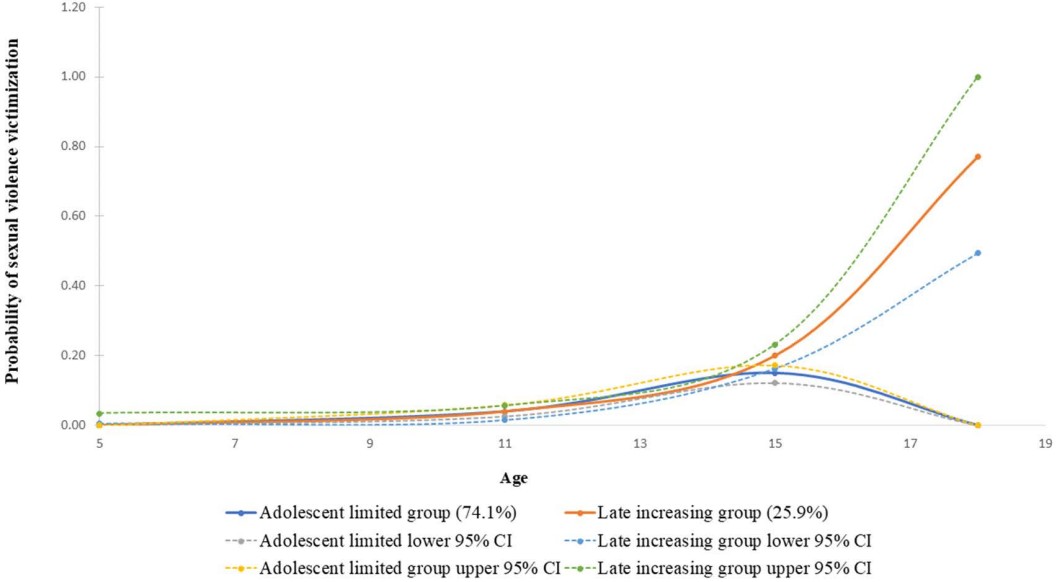

**Fig 3. Sexual violence victimization trajectories among children between 5 and 18 years of age.** Note; lines show estimated trajectories, points at each age represent the observed proportion of children between the age of 5 to 18 years reporting sexual violence victimization according to the assigned sexual violence trajectory group.

*adolescent limited* group was characterized by low sexual violence victimization at the age of five years, whose onset was at age 11 with a peak at age 15. This was followed by a decrease in experience of sexual violence from age 15. The *late increasing* trajectory group also showed onset at age 11, with sexual violence victimization persisting and increasing through to age 18. The model with a quadratic (2) and cubic (3) term provided the best description of sexual violence victimization trajectories. Although the 2-group model with quadratic terms had a slightly lower BIC compared to the model selected, the significantly lower entropy (0.44) for that model made it less desirable for describing sexual violence victimization patterns. The final model selected had average posterior probabilities of 0.85 and 1.00 for the *adolescent limited* and *late increasing* group, respectively, with an entropy of 0.57 (Table in S4 Table).

### Early life factors associated with physical and sexual violence victimization trajectories

Table 2 and 3 show results of the independent and adjusted associations between the early life factors and trajectory group membership for physical and sexual violence victimization, respectively. Both individual and family level factors were associated with physical violence victimization trajectories in the unadjusted and adjusted models. In contrast, none of the family level factors associated with sexual violence victimization trajectories remained statistically significant after controlling for other variables. The high proportion of missing data on certain variables (relative growth variables, maternal prior violence experience and mode of delivery) prevented their inclusion in the multivariable model.

**Factors associated with physical violence victimization trajectories.** Of the individual level factors, sex, relative infant weight gain and childhood height gain were independently associated with membership in the *chronic increasing* physical violence victimization trajectory group (Table 2). From the univariable analysis, an increase in infant weight was associated with 14% greater odds of membership in the *chronic increasing* trajectory group (OR 1.14, 95% CI 1.02; 1.27). The results show that height gain during childhood was protective against membership in the *chronic increasing* trajectory group. An increase in relative height was associated with 22% lower odds of membership in the *chronic increasing* trajectory group (OR 0.88, 95% CI 0.79; 0.99). After adjusting for other variables in the multivariable model, sex was the

**Table 2.  Univariable and multivariable logistic regression model showing early life factors associated with physical violence victimization trajectory group membership.**

| Variables | OR (95% CI) | P-values | aOR (95% CI) | P-values |
|---|---|---|---|---|
| **Individual level factors** | | | | |
| **Sex** | | <0.001 | | <0.001 |
| Female (ref) | 1.00 | | 1.00 | |
| Male | 1.77 (1.48; 2.13) | | 1.67 (1.31; 2.10) | |
| **Birth weight** | | 0.416 | | 0.479 |
| Normal (ref) | 1 | | 1 | |
| Low birth weight | 1.12 (0.85; 1.49) | | 1.15 (0.78; 1.68) | |
| **Infants and child growth** | | | | |
| Relative weight gain 0–2 years | 1.14 (1.02; 1.27) | 0.023 | | |
| Relative weight gain 2–5 years | 0.98 (0.87; 1,09) | 0.664 | | |
| Relative height gain 0–2 years | 0.96 (0.86; 1.07) | 0.486 | | |
| Relative height gain 2–5 years | 0.88 (0.79; 0.99) | 0.035 | | |
| **Family level factors** | | | | |
| **Household socioeconomic status** | | 0.006 | | 0.034 |
| Low (ref) | 1.00 | | 1.00 | |
| Middle | 0.71 (0.55; 0.90) | | 0.68 (0.50; 0.92) | |
| High | 0.73 (0.55; 0.98) | | 0.77 (0.54; 1.11) | |
| **Household crowding** | | 0.079 | | 0.909 |
| Yes (ref) | 1.00 | | 1.00 | |
| No | 0.84 (0.69; 1.02) | | 0.99 (0.77; 1.26) | |
| **Maternal age** | | | | |
| ≤ 24 years (ref) | 1.00 | 0.975 | 1.00 | 0.465 |
| 25–34 years | 0.99 (0.82; 1.20) | | 1.18 (0.87; 1.60) | |
| ≥ 35 years | 1.03 (0.75; 1.40) | | 1.29 (0.80; 2.08) | |
| **Parity** | | 0.395 | | 0.601 |
| One child (ref) | 1.00 | | 1.00 | |
| more than one child | 1.08 (0.90; 1.31) | | 0.92 (0.68; 1.25) | |
| **Marital Status** | | 0.096 | | 0.238 |
| Married (ref) | 1.00 | | 1.00 | |
| Single | 1.17 (0.97; 1.42) | | 1.18 (0.89; 1.57) | |
| **Maternal education status** | | 0.061 | | 0.041 |
| Primary & below | 1.63 (1.05; 2.53) | | 1.34 (0.74; 2.43) | |
| Secondary | 1.52 (1.06; 2.19) | | 1.73 (1.08; 2.76) | |
| Post-school training (ref) | 1.00 | | 1.00 | |
| **Paternal education status** | | 0.192 | | 0.869 |
| Primary & below | 1.26 (0.80; 1.99) | | 1.15 (0.66; 1.98) | |
| Secondary | 1.31 (0.98; 1.74) | | 1.08 (0.77; 1.51) | |
| Post-school training (ref) | 1.00 | | 1.00 | |
| **Father present** | | 0.473 | | 0.624 |
| Yes (ref) | 1.00 | | 1.00 | |
| No | 1.10 (0.85; 1.43) | | 1.10 (0.76; 1.59) | |
| **Maternal prior violence experience** | | 0.816 | | |
| Yes (ref) | 1.00 | | | |
| No | 1.04 (0.74; 1.46) | | | |
| **Mode of delivery** | | 0.082 | | |

*(Continued)*

**Table 2.** (Continued)

| Variables | OR (95% CI) | P-values | aOR (95% CI) | P-values |
|---|---|---|---|---|
| Vaginal (ref) | 1.00 | | | |
| Assisted | 0.70 (0.48; 1.04) | | | |

OR: odds Ratio. aOR: adjusted Odds Ratio. Ref: reference group. Adolescent limited violence trajectory group acts as the outcome reference group. Goodness of fit of the model; p = 0.262.

only individual level factor that was significantly associated with membership in the *chronic increasing* trajectory group. Boys had 67% higher odds of membership in the *chronic increasing* trajectory group compared to girls (aOR 1.67, 95% CI 1.31; 2.10).

Household socioeconomic status was the only family level factor significantly associated with membership in the *chronic increasing* physical violence victimization trajectory group, in the unadjusted analysis. In addition, the association between maternal education (p = 0.061), household crowding (p = 0.079) and mode of delivery (p = 0.082) approached statistical significance with *chronic increasing* trajectory group membership prior to adjusting for other factors. From the multivariable model, lower household socioeconomic status and lower maternal education were the only family level factors that predicted membership in the *chronic increasing* trajectory group, after adjusting for other variables in the model. Children from households within the middle socioeconomic status level had 32% lower odds of membership in the *chronic increasing* trajectory group compared to children residing in the low household socioeconomic status level (aOR 0.68, 95% CI 0.50; 0.92). Furthermore, children whose mothers had received at least some secondary education had 73% greater odds of *chronic increasing* trajectory group membership compared to mothers with post-school training (aOR 1.73, 95% CI 1.08; 2.76).

**Factors associated with sexual violence victimization trajectories.** Results from Table 3 demonstrate that only family level factors significantly predicted membership in the *late increasing* sexual violence victimization trajectory group in the univariable regression analysis (p < 0.05). Children from households with high socioeconomic status and no crowding had 42% (OR 0.58, 95% CI 0.41; 0.81) and 26% (OR 0.74, 95% CI 0.60; 0.92) lower odds of membership in the *late increasing* trajectory group compared to children from lower socioeconomic households and children from crowded households, respectively. Independently, the odds of membership in the *late increasing* trajectory group were greater with decreasing maternal (primary & below OR 2.56, 95% CI 1.55; 4.23, secondary OR 1.83, 95% CI 1.18; 2.83) and paternal (primary & below OR 1.70 95% CI 1.02; 2.83, secondary OR 1.60, 95% CI 1.19; 2.34) education compared to children from parents with post-school training. None of these factors remained statistically significant after adjusting for all other variables in the multivariable model. However, the association between household socioeconomic status and membership in the *late increasing* trajectory group approached significance (p = 0.07) in the multivariable analysis. Children residing in high socioeconomic households had 37% lower odds of membership in the *late increasing* trajectory group compared to children residing in households with low socioeconomic status (aOR 0.63, 95% CI 0.42; 0.95), after adjusting for covariates in the model.

Post regression goodness of fit analyses for the final models for the physical (p = 0.262) and sexual (p = 0.520) violence victimization trajectories, did not provide evidence of a lack of model adequacy. Low variance inflation factor (VIF) scores from the multivariable analyses – mean VIF of 1.2 for physical violence and 1.18 for sexual violence victimization – indicated no multicollinearity among covariates in the final models.

## Discussion

This study sought to examine the number and forms of physical and sexual violence victimization trajectories between the ages of five and 18 years, as well as identify factors in early life associated with trajectory group membership. The results

**Table 3. Univariable and multivariable logistic regression model showing early life factors associated with sexual violence victimization trajectory group membership.**

| Variables | OR (95% CI) | P-values | aOR (95% CI) | P-values |
|---|---|---|---|---|
| **Individual level factors** | | | | |
| **Sex** | | 0.574 | | 0.519 |
| Female (ref) | 1.00 | | 1.00 | |
| Male | 1.06 (0.87; 1.29) | | 0.92 (0.71; 1.19) | |
| **Birthweight** | | 0.312 | | 0.848 |
| Normal (ref) | 1.00 | | 1.00 | |
| Low birth weight | 1.17 (0.86; 1.59) | | 1.04 (0.69; 1.57) | |
| **Infant and child growth** | | | | |
| Relative weight gain 0–2 years | 1.02 (0.90; 1.15) | 0.736 | | |
| Relative weight gain 2–5 years | 0.99 (0.88; 1.12) | 0.915 | | |
| Relative height gain 0–2 years | 0.95 (0.84; 1.07) | 0.393 | | |
| Relative height gain 2–5 years | 0.91 (0.80; 1.03) | 0.128 | | |
| **Family level factors** | | | | |
| **Household socioeconomic status** | | <0.003 | | 0.0671 |
| Low (ref) | 1.00 | | 1.00 | |
| Middle | 0.81 (0.62; 1.05) | | 0.82 (0.59; 1.13) | |
| High | 0.58 (0.41; 0.81) | | 0.63 (0.42; 0.95) | |
| **Household crowding** | | 0.007 | | 0.300 |
| Yes (ref) | 1.00 | | 1.00 | |
| No | 0.74 (0.60; 0.92) | | 0.87 (0.66; 1.13) | |
| **Maternal age** | | 0.248 | | 0.862 |
| ≤ 24 years (ref) | 1.00 | | 1.00 | |
| 25–34 years | 0.91 (0.74; 1.13) | | 0.93 (0.67; 1.29) | |
| ≥ 35 years | 1.20 (0.87; 1.67) | | 1.02 (0.61; 1.72) | |
| **Parity** | | 0.332 | | 0.721 |
| One child (ref) | 1.00 | | 1.00 | |
| more than one child | 1.11 (0.90; 1.36) | | 1.06 (0.76; 1.47) | |
| **Marital Status** | | 0.761 | | 0.768 |
| Married (ref) | 1.00 | | 1.00 | |
| Single | 1.03 (0.84; 1.27) | | 1.05 (0.77; 1.42) | |
| **Maternal education status** | | 0.001 | | 0.327 |
| Primary & below | 2.56 (1.55; 4.23) | | 1.63 (0.85; 3.13) | |
| Secondary | 1.83 (1.18; 2.83) | | 1.31 (0.77; 2.23) | |
| Post-school training (ref) | 1.00 | | 1.00 | |
| **Paternal education status** | | 0.012 | | 0.337 |
| Primary & below | 1.70 (1.02; 2.83) | | 1.13 (0.62; 2.06) | |
| Secondary | 1.67 (1.19; 2.34) | | 1.32 (0.89; 1.94) | |
| Post-school training (ref) | 1.00 | | 1.00 | |
| **Father present** | | 0.741 | | 0.225 |
| Yes (ref) | 1.00 | | 1.00 | |
| No | 0.95 (0.71; 1.27) | | 0.77 (0.50; 1.18) | |
| **Maternal prior violence experience** | | 0.432 | | |
| Yes (ref) | 1.00 | | | |
| No | 1.17 (0.80; 1.71) | | | |
| **Mode of delivery** | | 0.482 | | |

*(Continued)*

**Table 3.** (Continued)

| Variables | OR (95% CI) | P-values | aOR (95% CI) | P-values |
|---|---|---|---|---|
| Vaginal (ref) | 1.00 | | | |
| Assisted | 0.86 (0.56; 1.31) | | | |

OR: Odds Ratio. aOR: adjusted Odds Ratio. Ref: reference group. Adolescent limited violence trajectory group acts as the outcome reference group. Goodness of fit of the model; p=0.520

from the group-based trajectory modelling identified two trajectories of physical and two trajectories of sexual violence victimization for the best description of physical violence victimization and sexual violence victimization patterns. Overall, both the individual level – sex – and family level – household socioeconomic status and maternal education – factors were identified as important determinants of physical violence victimization trajectory group membership. However, vulnerability to increasing sexual violence victimization was largely determined by family level factors, notably household socioeconomic status.

For patterns of physical violence, the findings show that the majority of the children in the present study exhibited an increasing pattern of physical violence as they grew older and engaged with more environments outside the home. These experiences of physical violence victimization primarily occurred between early childhood and mid-adolescence. Previous literature supports these findings showing that cases of physical violence victimization peak during early adolescent years and decrease with increasing age after mid-adolescence [40–42]. However, for a third of the children in this study, their experience of physical violence did not decrease in mid-adolescence, and gradually increased as they reached adulthood. Evidence of the presence of an *adolescent limited* and *increasing* physical violence trajectory group was reported by Semenza et al. [11], among individuals assessed across four data collection points between 12 and 34 years of age. Similar to our findings, the *adolescent limited* trajectory group was characterized by elevated levels of violence victimization in early adolescence that decreased rapidly as respondents transitioned to adulthood. In contrast, the *increasing* physical violence trajectory group was characterized by low experiences of physical violence in adolescence, which then increased into adulthood. Two other trajectory groups, *little to no victimization* and *high decreasing* trajectory groups were also identified by these authors [11].

Experiences of sexual violence victimization were concentrated in the adolescent period. Parent-reported cases of sexual violence victimization were less than 1% for children at five years of age. This was followed by a steady increase in experiences of sexual violence with age during adolescence, and by the age of 18, nearly one in three children in the sample experienced some form of sexual violence. Other studies have documented a higher likelihood of sexual violence victimization among adolescent and adult samples compared to younger child samples [41,43]. For three quarters of the children, low levels of sexual violence experiences were reported at 11 years, with experience of sexual violence decreasing from 15 years of age. In contrast, a quarter of these children's experience of sexual violence increased exponentially as they approached adulthood. Similar to our results, Jones et al. [9] generated two trajectory groups of sexual violence victimization assessed between the ages of 2–12 years. The two groups were (a) no sexual violence victimization, (b) a curvilinear pattern of sexual violence victimization characterized by low violence allegations at younger and older ages and a peak in sexual violence victimization between ages 4–8 years. This study recruited high risk children with reported alleged or substantiated allegations of child maltreatment or witnessing violence from child protective services records, which may exclude those in the population not within this radar [9].

Our findings show that certain individual level factors predicted *chronic increasing* physical violence victimization trajectories. Independent of other factors, boys were almost twice as likely as girls to be in the *chronic increasing* trajectory group. Similar findings have been demonstrated in other studies [2,11,40,44], and a possible reason for this is the fact that boys experience more conduct problems and exhibit greater externalizing behaviour compared to girls, which places

them at a heightened risk for both physical violence victimization and perpetration [20,45]. Weight and height gain in early life were associated with physical violence victimization trajectories. From the univariable analyses, an increase in infant weight was a risk factor for *chronic increasing* physical violence trajectory group membership. In addition, being taller had a protective effect against membership in the *chronic increasing* physical violence victimization group. Poor linear growth (stunting) during infancy has been associated with delayed cognitive development, poor educational outcomes in childhood [29,31], increased weight gain and obesity in later childhood, adolescence and adulthood [29,46] – all factors that subsequently increase the risk of violence victimization [33]. Moreover, previous research has shown that childhood stunting results from the complex interplay between demographic, cultural, economic and environmental factors [47,48]. These same factors are linked to violence victimization, suggesting a need for further research on the association between childhood stunting and VAC, taking into account these underlying influences.

None of the individual-level factors were significantly associated with sexual violence victimization trajectories in either the crude or adjusted analyses. Although boys had a lower likelihood of membership in the *late increasing* sexual violence victimization group, this association was not significant. Contrary to this, many studies report significantly higher prevalence of sexual violence among girls than boys [41,43,49]. The lack of a significant difference in trajectory group membership between boys and girls may be explained by the use of self-completed questionnaires from the age of 11 years, which has been documented to contribute to higher disclosure on sexual violence experiences, especially among boys [50]. None of the infant and child growth factors were associated with *late increasing* sexual violence victimization group membership. Richter et al. [51] found conflicting results in the same cohort with stunted boys having a higher likelihood of sexual violence experience compared to boys with normal height for their age.

Family level factors – household socioeconomic status and maternal education – independently, and when adjusted for other confounding variables, predicted physical violence victimization trajectory group membership. Adjusting for other factors, higher household socioeconomic status was associated with a lower risk of membership in the *chronic increasing* physical violence victimization trajectory group. This corresponds with evidence from other studies showing that physical violence experiences are higher among poorer households [33,44]. Furthermore, maternal education was found to be protective against *chronic increasing* physical violence victimization trajectory group membership. Societal gender roles assign the responsibility of raising children to mothers in the household, putting further emphasis on the role of women in children's exposure and experiences of violence. A study by Sui et al. [13], demonstrated that maternal presence in the household was protective in terms of both violence victimization and perpetration. Similar to our results, Semenza et al. [11], found that higher parental education was associated with a lower probability of membership in the increasing physical violence victimization compared to no victimization trajectory group. However, no association was observed between paternal education – a variable associated with household financial support – and *chronic increasing* physical violence victimization trajectories in this current study. The spillover effects of post-Apartheid migration policies on family disruption, low and delayed marriage rates and high unemployment rates have all contributed to increased rates of paternal absenteeism, non-residence or non-involvement in South African households [52]. Bitalo et al. [53] argued that the role of fathers should not be limited to the biological association or financial provision alone, rather more emphasis should be placed on the father's or father figure's active presence and involvement. The lack of this paternal involvement is a documented risk factor for multiple types of violence [23].

For sexual violence victimization trajectories, higher socioeconomic status appeared to be protective against inclusion in the *late increasing* sexual violence victimization group. However, this association was not significant after other factors were taken into account. Richter et al. [51] found similar findings, with household socioeconomic status being only significantly associated with sexual abuse among boys prior to adjusting for other covariates. Other family level factors associated with *late increasing* sexual violence victimization trajectories, only in the unadjusted analysis, include lower maternal and paternal education and household crowding. Consistent with our findings, Ward et al. [50], found an association that approached significance in the univariable, but not in the multivariable analysis, between sharing a bedroom with more

than one person and sexual abuse. Similar to our findings, marital status, maternal education and father's presence were not found to be significantly associated with sexual abuse among boys in South Africa [51].

South Africa's context of high levels of all forms of violence and multiple adversities mean that many children are often raised in contexts of high adversity experienced as normative, and indeed moderate levels of adversity have been shown to contribute to resilience [54]. Further research is needed to fully understand the development of resilience in high adversity settings, and its protective role against victimization group inclusion, to expand some of the important contextualized resilience research being undertaken in South Africa. Notably, Theron [55] posits that the development of resilience in response to significant adversity must involve interplay between individuals and their social systems as an adaptive and dynamic process. And in her work, Theron finds that youth age, caregiver warmth, school resources and teacher competence all contribute to South African learners' levels of school engagement, as a function of resilience [56]. Financial security in the home, positive parenting and the health of the caregiver have been found to be protective of children's violence victimization [33]. Much more work needs to be done for a comprehensive understanding of the development of resilience from early childhood through adolescence, and its role along the life course in predicting children's victimization.

Interpretation of the study findings should take into consideration several important limitations. First, the use of secondary data limited control over the timing of key measurements. In particular, violence victimization data were not collected at consistent intervals across childhood and adolescence, which may have limited the capacity to detect more nuanced or heterogeneous trajectories of violence over time. Furthermore, the dataset did not include all potentially relevant early life exposures, and completeness of data for certain variables was a concern in some cases. This possibly contributed to residual confounding, which persists despite final model demonstrating adequate fit. Second, the use of complete case analyses raises concerns about potential selection bias. Participants with missing data on key exposures or outcomes were excluded from the analysis, which may have resulted in a sample that is systematically different from those lost to follow-up or excluded due to incomplete data. Although efforts were made to compare included and excluded samples, the possibility remains that observed associations were influenced by this attrition.

Third, the reliance on caregiver- and self-reported measures of violence exposure, particularly for sensitive issues such as sexual violence, may have introduced social desirability or reporting bias. Self-completed questionnaires from age 11 likely improved disclosure, especially among boys, but underreporting—particularly in early childhood when caregivers serve as proxy respondents—remains a concern. The stigma surrounding sexual and physical abuse, as well as normative beliefs about corporal punishment, may have further influenced response patterns. Fourth, the study population was drawn from a predominantly Black, peri-urban South African cohort. While this provides critical insight into violence trajectories in a historically marginalized and under-researched context, it also limits generalizability to other population groups within South Africa and beyond. Structural factors such as legacies of apartheid spatial planning, migration, and economic exclusion shape this context in unique ways.

Lastly, although GBTM is a powerful method for identifying distinct developmental patterns, trajectory modelling is inherently exploratory and can be influenced by model specification choices, including the number of groups and polynomial orders. While the average posterior probabilities were high, indicating good classification of individuals into trajectory subgroups, the moderate entropy values suggest some uncertainty in class assignment. This uncertainty can influence the precision of risk factor associations with trajectory group membership. Future research should incorporate additional measurement waves to improve trajectory resolution, explore alternative clustering methods, and use a broader set of model adequacy indices, such as the parametric bootstrapped likelihood ratio test (BLRT) and Lo-Mendell-Rubin test (LMR-LRT), to strengthen confidence in trajectory classifications.

Despite these limitations, the study's prospective design, large sample size, and application of life course-informed statistical methods offer several strengths. Unlike retrospective studies that are vulnerable to long recall periods, this study used repeated measures across key developmental stages, helping to minimize recall bias and better capture the timing and progression of violence experiences. Furthermore, the application of trajectory modelling contributes to an emerging literature

on longitudinal patterns of violence victimization in African contexts, where evidence remains sparse. Finally, the study's integration of both individual and family-level early life exposures aligns with the developmental origins of health and disease perspective. While DOHaD research has traditionally focused on physical health outcomes, these findings suggest that early nutritional and socioeconomic conditions may also shape vulnerability to psychosocial and interpersonal harms, such as violence victimization. This broadens the application of the DOHaD perspective to include developmental pathways of risk and resilience in the social domain, offering new insights for early intervention and prevention strategies in high-adversity settings.

## Conclusion

This study demonstrates that even within shared neighbourhoods and environments, children can follow markedly different developmental patterns of violence victimization. Using group-based trajectory modelling in a longitudinal South African cohort, we identified distinct pathways of both physical and sexual violence from early childhood through adolescence. This approach offers a novel contribution to the violence literature in African contexts, where prospective studies of this kind remain rare.

The analysis revealed two key patterns of physical violence victimization: a large group experiencing increasing exposure through adolescence, and a smaller group with persistent and escalating experiences. Individual-level factors, sex and early growth indicators such as higher infant weight and shorter stature, were associated with greater risk of belonging to the chronic increasing trajectory. Family-level determinants, including lower household socioeconomic status and maternal education, also emerged as significant and consistent predictors.

In contrast, sexual violence victimization trajectories were more sharply concentrated in adolescence. While some children experienced only transient or declining exposure, a notable subgroup faced sharply increasing risk as they neared adulthood. Unlike physical violence, no individual-level factors were significantly associated with these trajectories in adjusted models, though family-level socioeconomic adversity showed some influence in unadjusted analyses.

These findings have important implications for the timing and design of early interventions. First, they highlight the value of integrated, multisectoral programmes that combine poverty alleviation with parenting and nutritional support, especially during the first 1,000 days of life. Addressing early stunting and promoting healthy development may have long-term protective effects. Second, households with low maternal education and fewer resources should be prioritized for targeted violence prevention efforts. However, the heterogeneity of trajectories also points to the need for universal, population-level strategies that can reach vulnerable children who do not fall into easily defined "high-risk" groups. A hybrid model that combines targeted interventions for chronically exposed children with broader structural initiatives may be the most effective and equitable approach, particularly in the face of persistent inequality, poverty, and systemic barriers. Community-led and contextually grounded solutions will be vital to ensuring feasibility and sustainability in this context.

In conclusion, this study contributes new evidence on developmental patterns of violence victimization and their early life origins in South Africa. It highlights the importance of longitudinal approaches for identifying risk and resilience pathways, and calls for investment in early, sustained, and layered intervention strategies that span the individual, household, and structural levels. Future research should build on this work by exploring mechanisms of resilience, refining early risk markers, and testing the effectiveness of intervention models tailored to these developmental trajectories.

## Supporting information

**S1 Table. Proportions of physical and sexual violence victimization per age.**
(DOCX)

**S2 Table. Characteristics of excluded and included sample for physical violence victimization trajectory analyses.**
(DOCX)

**S3 Table. Characteristics of included and excluded sample for sexual violence victimization trajectory analyses.**
(DOCX)

**S4 Table. Physical and sexual violence victimization trajectory model selection and adequacy.**
(DOCX)

**S5 Text. Sample power calculation.**
(DOCX)

## Acknowledgments

The authors would like to thank the Birth to Twenty-Plus participants for their role in the study, as well as the Developmental Pathways to Health Research Unit at the University of the Witwatersrand where the study is housed.

## Author contributions

**Conceptualization:** Lilian Muchai, Sara Naicker, Juliana Kagura.

**Data curation:** Lilian Muchai.

**Formal analysis:** Lilian Muchai, Sara Naicker, Juliana Kagura.

**Funding acquisition:** Sara Naicker, Juliana Kagura.

**Methodology:** Lilian Muchai, Sara Naicker, Juliana Kagura.

**Project administration:** Lilian Muchai, Sara Naicker, Juliana Kagura.

**Supervision:** Sara Naicker, Juliana Kagura.

**Validation:** Lilian Muchai, Sara Naicker, Juliana Kagura.

**Visualization:** Lilian Muchai, Sara Naicker, Juliana Kagura.

**Writing – original draft:** Lilian Muchai.

**Writing – review & editing:** Lilian Muchai, Sara Naicker, Juliana Kagura.

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
