## [Decision Letter · Decision Letter 0]

10 Jul 2024

Dear Dr. Muchai,

Thank you for submitting your manuscript to 29 July. After careful consideration, we feel that it has merit but does not fully meet PLOS ONE’s publication criteria as it currently stands. Therefore, we invite you to submit a revised version of the manuscript that addresses the points raised during the review process.

The paper includes many additional English language errors. Editing by an English language expert is needed and would significantly improve the article.

To strengthen the takeaway, I would recommend expanding on the promising areas and actionable strategies for early intervention in order to apply the knowledge gained from your manuscript in real-world contexts.

We look forward to receiving your revised manuscript.

Kind regards,

Giuseppe Marano

Academic Editor

PLOS ONE

 [The Bt20 Plus study is funded by the Wellcome Trust (UK), the Medical Research Council of South Africa, the University of Witwatersrand and supported by the DST - NRF Centre of Excellence in Human Development at the University of Witwatersrand, Johannesburg. The funders had no role in the study design, analysis, decision to publish or preparation of the manuscript.].  

4. In the online submission form you indicate that your data is not available for proprietary reasons and have provided a contact point for accessing this data. Please note that your current contact point is a co-author on this manuscript. According to our Data Policy, the contact point must not be an author on the manuscript and must be an institutional contact, ideally not an individual. Please revise your data statement to a non-author institutional point of contact, such as a data access or ethics committee, and send this to us via return email. Please also include contact information for the third party organization, and please include the full citation of where the data can be found.

Additional Editor Comments (if provided):

Reviewers' comments:

Reviewer's Responses to Questions

**Comments to the Author**

1. Is the manuscript technically sound, and do the data support the conclusions?

Reviewer #1: Yes

Reviewer #2: Yes

Reviewer #3: Yes

2. Has the statistical analysis been performed appropriately and rigorously?

Reviewer #1: Yes

Reviewer #2: Yes

Reviewer #3: Yes

3. Have the authors made all data underlying the findings in their manuscript fully available?

Reviewer #1: Yes

Reviewer #2: No

Reviewer #3: No

4. Is the manuscript presented in an intelligible fashion and written in standard English?

Reviewer #1: Yes

Reviewer #2: Yes

Reviewer #3: Yes

Reviewer #1: The present study, entitled Early life factors associated with childhood trajectories of violence among the Birth to Twenty-Plus Cohort in Soweto, South Africa, deals with the analysis of data from a database specifically on the issue of violence in children aged 5 to 18 years. It is an important, current and relevant study in the context of child and adolescent care in developing countries.

The authors use the appropriate scientific technical language, objective and pertinent to the theme, although it can be summarized, in order to allow a fluid and direct reading, especially in the descriptions of results. It should be noted that in these aspects, tables are important elements of information to the reader.

The authors employ diverse, pertinent references, but with more than 5 years for the most part. This aspect can weaken the discussions specifically on this topic in the work.

The method is described clearly and objectively in order to allow the reader to understand it and the possibility of replicability is guaranteed. The reviewer takes the liberty of suggesting to the authors the presence of a safe, infographic, which can summarize the flow of the process and allow the reader a better understanding of the sequence of steps.

The results presented are compatible with the objectives explained and resulting from the defined methodology. Sometimes the reader is faced with an amount of information that can compromise his understanding, to which the reviewer once again takes the liberty of suggesting, a presentation with the use of topics that would lead to the results, thus minimizing any possibility of confusion of the data presented.

The authors present results in tables and graphs, but it should be noted that the tables do not have margins or lines.

The suggestions presented here can quickly be rectified by the authors, impacting the short investment time. However, these rectifications are very important for a better quality presentation of the work. Since it is an important study in a context of violence and exposure to violence, the manuscript will certainly be very well appreciated by the scientific community.

Reviewer #2: I appreciate the opportunity to review this interesting manuscript. The focus of the study is well-defined, and the originality of the research is unquestionable as it differs from other literature. The study also shows adequate knowledge and understanding of relevant literature on the concept. The variables used in the study are well-defined. The discussion section is well written, with the authors making adequate references to other studies related to the outcome of their study. Additionally, the sentences are clearly expressed and readable. This is a well-written manuscript, and I look forward to seeing it in print. Thank you.

Reviewer #3: This manuscript presents interesting data on violence trajectories from the Birth to Twenty-Plus cohort in South Africa. Overall I found the manuscript to be clearly written, and my suggestions for improvement are relatively minor, and are mainly for the discussion, which I think needs some tightening up.

1. I would suggest avoiding the terms ‘developed’ and ‘developing’ when describing countries, and would recommend consulting the following reference for some helpful insights on terminology:

1.

Khan T, Abimbola S, Kyobutungi C, Pai M. How we classify countries and people—and why it matters. BMJ Glob Health. 2022 Jun;7(6):e009704.

2. Page 5, line 109: I think ‘covering’ should be removed, or remove the brackets around the years.

3. Page 22, lines 390-391: This sentence seems repetitive of what was discussed in the previous paragraph. I would suggest consolidating these to avoid repetition.

4. Page 23, lines 409-412: The comment about perpetrators, while relevant to the discussion, seems out of place in this paragraph, given that the following paragraph returns to the discussion of the age at which sexual violence is more prevalent.

5. Page 24, last paragraph: Given the prevalence of stunting in South Africa, and its association (not only in SA) with poor developmental outcomes, I think some discussion about the relationship between stunting and other vulnerabilities associated with violence would be helpful. The comment about overweight/obesity being a risk factors for bullying and teasing in childhood needs to be linked to the overall discussion about violence in this context. The last sentence of this paragraph (on page 25) would fit better earlier in the paragraph.

6. Page 26, lines 464-466 and 470-472: The research cited needs to be further expanded on to better link to the current study’s findings and context. These make sense to me, as someone working in the South African context, but may not make sense to an international reader unfamiliar with the context.

7. Page 27: The conclusion feels like it’s missing some recommendations for this context (and other similar contexts), based on the findings. It would be helpful to expand on the potential early target prevention areas mentioned.

**Do you want your identity to be public for this peer review?** For information about this choice, including consent withdrawal, please see our Privacy Policy

Reviewer #1: **Yes: ** JOAO CARLOS ALCHIERI

Reviewer #2: No

Reviewer #3: No

---

## [Author Response · Author response to Decision Letter 1]

23 Aug 2024

Dear Dr. Giuseppe Marano

The authors appreciate all the constructive feedback provided by the academic editor and the reviewers. The following documents requested for in your email dated July 10 have been uploaded in the online submission system:

1. A rebuttal letter with responses to each of the reviewers' comments (Response to Reviewers).

2. A marked-up copy of the manuscript (Revised Manuscript with Track Changes).

3. An unmarked version without track changes (Manuscript)

The following revision to the data availability statement has also been made, removing co-authors as contact points for data access request in line with the journal Data Policy:

The Birth to Twenty- Plus Cohort data can be requested from the Birth to Thirty Executive Committee: Prof. Linda Ritcher, the executive committee chair, linda.richter@wits.ac.za. The first five years of cohort data are available online at bt30.org (https://bt30.org/data/ ).

Thank you for your time and we look forward to your favorable response.

Lilian Muchai

---

## [Decision Letter · Decision Letter 1]

30 Oct 2024

Dear Dr. Muchai,

Thank you for submitting your manuscript to PLOS ONE. After careful consideration, we feel that it has merit but does not fully meet PLOS ONE’s publication criteria as it currently stands. Therefore, we invite you to submit a revised version of the manuscript that addresses the points raised during the review process.

We look forward to receiving your revised manuscript.

Kind regards,

Giuseppe Marano

Academic Editor

PLOS ONE

Journal Requirements:

Reviewers' comments:

Reviewer's Responses to Questions

**Comments to the Author**

Reviewer #4: (No Response)

Reviewer #5: All comments have been addressed

2. Is the manuscript technically sound, and do the data support the conclusions?

Reviewer #4: Yes

Reviewer #5: Yes

3. Has the statistical analysis been performed appropriately and rigorously?

Reviewer #4: I Don't Know

Reviewer #5: Yes

4. Have the authors made all data underlying the findings in their manuscript fully available?

Reviewer #4: Yes

Reviewer #5: Yes

5. Is the manuscript presented in an intelligible fashion and written in standard English?

Reviewer #4: Yes

Reviewer #5: Yes

Reviewer #4: The subject matter and content of this research is very interesting. A good amount of the mild hesitation I have about the manuscript is primarily of an overarching/general nature, but there are a few comments on specific suggestions as well.

SPECIFIC COMMENTS/OBSERVATIONS:

-any findings with significance values of greater than 0.05 should likely not be termed "marginally significant" but rather "approaches significance" or "trend-level" or something to similar effect as from a term-of art standpoint a p>0.05 would not be "significant."

-when there is an overwhelming amount of results data, it is helpful to provide organization for the reader in the Discussion section. Providing some structure in the Discussion about different categories/groupings of risk factors (by individual-level factor, family-level factor) or organizing the Discussion by victimization trajectory would provide a framework to provide clearer speculative explanations about potential data meaning in terms of interpretation as well as what it might implicate for next steps in research +/- intervention.

GENERAL COMMENTS/OBSERVATIONS:

On page 26 of the manuscript there is a sentence that reads, "The spillover effects of post-Apartheid migration policies on family disruption, low marriage rates, alcohol and substance abuse and financial difficulties fathers face in providing for their families, have all contributed to increased rates of paternal absenteeism, non-residence or non-involvement in South African households [51]." While this general comment may have been highlighting a specific issue re: family structure factors, it speaks to some of the general impacts of societal stress that affect much of South African society--as the authors allude to at various points in the manuscript. If this is the case, it likely behooves the authors to do more to explain what protects youth from victimization group assignment. Said alternately: Are the risk factors *per se* predictive of assignment to victimization trajectories, or is it an interaction between a predictive factor and a wider social milieu?

The authors also highlight stunted development as something that is affects almost 30% of the overall population. After this study, an imagined intervention purportedly focused on children with stunted growth would most likely be a generalized primary prevention strategy rather than focused on a small population subset. There likely needs to be a clearer discussion not just of risk specificity of predictive factors, but also on whether the likely subsequent intervention strategies should be targeted vs. generalized across the entire youth population. Wrestling with the wider implications of this research and how it likely interacts with general social characteristics in the country would help to better contextualize how to interpret these results.

Reviewer #5: I would like to thank you authors for addressing all previous comments so thoroughly. This is a well-written manuscript, with a robust analysis and sound conclusions. I have very minor comments:

Line 271: The figure 2 legend has been mislabelled.

Line 290: The figure 3 legend has been mislabelled.

**Do you want your identity to be public for this peer review?** For information about this choice, including consent withdrawal, please see our Privacy Policy

Reviewer #4: No

Reviewer #5: No

---

## [Author Response · Author response to Decision Letter 2]

13 Dec 2024

Dear Dr. Giuseppe Marano,

The authors of the manuscript titled “Early life factors associated with childhood trajectories of violence among the Birth to Twenty-Plus Cohort in Soweto, South Africa,” appreciate the time and effort taken by the reviewers in re-evaluating our work. All the comments were carefully considered, and corresponding revisions were made to enhance the quality and clarity of the manuscript.

The following documents requested for in your email dated October 30 have been uploaded in the online submission system:

1. A rebuttal letter with responses to each of the reviewers' comments (Response to Reviewers).

2. A marked-up copy of the manuscript (Revised Manuscript with Track Changes).

3. An unmarked version without track changes (Manuscript).

Thank you for your time and we look forward to your favorable response.

Sincerely,

Lilian Muchai

---

## [Decision Letter · Decision Letter 2]

8 Jul 2025

Dear Dr. Muchai,

Thank you for submitting your manuscript to PLOS ONE. After careful consideration, we feel that it has merit but does not fully meet PLOS ONE’s publication criteria as it currently stands. Therefore, we invite you to submit a revised version of the manuscript that addresses the points raised during the review process.

I would like to ask you to carefully follow all the suggestions provided by the reviewers. In particular, I encourage you to strengthen the Discussion section by offering a more in-depth reflection on the limitations of the study. I also invite you to revise and expand the Conclusions, especially with regard to the broader implications of your findings. 

Please submit your revised manuscript by the Aug 22 2025 11:59PM If you will need more time than this to complete your revisions, please reply to this message or contact the journal office at plosone@plos.org . A rebuttal letter that responds to each point raised by the academic editor and reviewer(s). You should upload this letter as a separate file labeled 'Response to Reviewers'.A marked-up copy of your manuscript that highlights changes made to the original version. You should upload this as a separate file labeled 'Revised Manuscript with Track Changes'.An unmarked version of your revised paper without tracked changes. You should upload this as a separate file labeled 'Manuscript'.

We look forward to receiving your revised manuscript.

Kind regards,

Giuseppe Marano

Academic Editor

PLOS ONE

Journal Requirements:

Reviewers' comments:

Reviewer's Responses to Questions

**Comments to the Author**

Reviewer #6: All comments have been addressed

Reviewer #7: (No Response)

2. Is the manuscript technically sound, and do the data support the conclusions?

Reviewer #6: Partly

Reviewer #7: Yes

3. Has the statistical analysis been performed appropriately and rigorously?

Reviewer #6: No

Reviewer #7: Yes

4. Have the authors made all data underlying the findings in their manuscript fully available?

Reviewer #6: No

Reviewer #7: Yes

5. Is the manuscript presented in an intelligible fashion and written in standard English?

Reviewer #6: No

Reviewer #7: Yes

Reviewer #6: While the limitations section addresses some key methodological concerns, it would benefit from additional clarification. Specifically, the use of secondary data limits control over the timing of measurements and the availability of certain variables, which should be more explicitly acknowledged. It is also important to note that the approach to missing data could introduce selection bias, and I recommend that future studies conduct sensitivity analyses to assess the robustness of trajectory groupings. Furthermore, although model diagnostics were conducted, residual confounding remains a possibility and should be highlighted more explicitly.

Given that this study uses secondary data from an existing longitudinal cohort, there are limitations in incorporating certain potentially relevant factors (such as maternal prior violence, mode of delivery, and infant growth) into the analyses due to high levels of missingness. While adjustments were made for variables that differed between included and excluded samples, selection bias cannot be fully ruled out. Notably, children included in the trajectory analyses were more likely to come from lower socioeconomic households with lower parental education, which may have influenced the observed associations. Additionally, the classification into trajectory groups was based on model fit indices (e.g., BIC, posterior probabilities, entropy), but the entropy values for both physical and sexual violence models were moderate (around 0.56–0.57), indicating some uncertainty in group assignment. This could attenuate the strength of associations between early life predictors and trajectory group membership. Lastly, while covariates were tested in multivariable models, the possibility of residual confounding remains, particularly due to the exclusion of relevant variables and potential multicollinearity among socioeconomic indicators.

Regarding the conclusion, while it does reflect the main findings of the study, it would benefit from greater clarity and structure. Specifically, it would be helpful to more clearly distinguish between the patterns and associated factors of physical versus sexual violence victimization. Additionally, I suggest emphasizing the novelty of using trajectory modeling in a longitudinal cohort more strongly. The limitations section currently occupies a disproportionate amount of space and could be summarized more succinctly to maintain focus on the study’s contributions. Finally, the conclusion would be more impactful with a concise summary statement that clearly articulates the implications for future research and intervention strategies.

In the Methods section, I recommend providing a clearer distinction between the two-step analytical process—presenting univariable logistic regressions first, followed by multivariable modeling. Statistical terminology should also be refined for conceptual accuracy. For example, the term "correlates" (or "correlation") should be replaced with "associations," as the analyses assess relationships rather than statistical correlations. It would also be helpful to clarify that the variables included in the multivariable model were selected based on prior literature and data availability. Additionally, post-regression diagnostics should be described more explicitly, including a note that a test for multicollinearity was conducted to assess interdependence among explanatory variables. Finally, some minor edits are needed to improve clarity, precision, and consistency throughout the manuscript.

Reviewer #7: I can see across revisions that the authors have made significant improvements, and the manuscript overall has become quite tidy. It seems to pass all guidelines for dual publication, research ethics, legibility, data availability, etc. With minor revisions related to clarity, as well as tying up a couple loose ends in the methodology, this will be suitable for publication.

**Do you want your identity to be public for this peer review?** For information about this choice, including consent withdrawal, please see our Privacy Policy

Reviewer #6: **Yes: ** Marina Xavier Carpena

Reviewer #7: No

---

## [Author Response · Author response to Decision Letter 3]

22 Aug 2025

Dear Dr. Marano

The authors appreciate the time and effort taken by the reviewers in re-evaluating our work. The authors have strengthened the discussion section by providing a more in-depth consideration of the study limitations and have also reinforced the conclusion regarding the implications of the findings. Individual reviewer comments were carefully addressed, and corresponding revisions were made to improve the quality and clarity of the manuscript.

Regards,

Lilian Muchai

---

## [Decision Letter · Decision Letter 3]

30 Oct 2025

Early life factors associated with childhood trajectories of violence among the Birth to Twenty-Plus Cohort in Soweto, South Africa.

PONE-D-23-32616R3

Dear Dr. Muchai,

We’re pleased to inform you that your manuscript has been judged scientifically suitable for publication and will be formally accepted for publication once it meets all outstanding technical requirements.

Kind regards,

Giuseppe Marano

Academic Editor

PLOS ONE

Additional Editor Comments (optional):

Reviewers' comments:

Reviewer's Responses to Questions

**Comments to the Author**

Reviewer #7: All comments have been addressed

Reviewer #8: All comments have been addressed

2. Is the manuscript technically sound, and do the data support the conclusions?

Reviewer #7: Yes

Reviewer #8: Yes

3. Has the statistical analysis been performed appropriately and rigorously?

Reviewer #7: Yes

Reviewer #8: Yes

4. Have the authors made all data underlying the findings in their manuscript fully available?

Reviewer #7: Yes

Reviewer #8: Yes

5. Is the manuscript presented in an intelligible fashion and written in standard English?

Reviewer #7: Yes

Reviewer #8: Yes

Reviewer #7: It is evident that the authors wrote with care and precision, offering clearly-articulated and insightful interpretations of the findings. No paper is perfect, yet any minor tweaks that could still be made in this one would not affect the overall quality, nor the ability for other scholars to gain meaningful information from the published article. Thus, I recommend this draft for acceptance.

Reviewer #8: The authors have satisfactorily addressed the questions posed in the initial submission. The manuscript is now more thorough and easier to read. It is now more appropriate technically. The manuscript may now contribute to the body of scientific literature in its current form.

Well done. No further modifications are needed.

**Do you want your identity to be public for this peer review?** For information about this choice, including consent withdrawal, please see our Privacy Policy

Reviewer #7: No

Reviewer #8: **Yes: ** Gyanesh Kumar Tiwari

---

## [Editor Report · Acceptance letter]

PONE-D-23-32616R3

PLOS ONE

Dear Dr. Muchai,

I'm pleased to inform you that your manuscript has been deemed suitable for publication in PLOS ONE. Congratulations! Your manuscript is now being handed over to our production team.

Kind regards,

on behalf of

Dr. Giuseppe Marano

Academic Editor

PLOS ONE